# Application of Encapsulation Technology: In Vitro Screening of Two *Ficus carica* L. Genotypes under Different NaCl Concentrations

Irene Granata [1],*[iD], Luca Regni [2][iD], Maurizio Micheli [2],*[iD], Cristian Silvestri [3][iD] and Maria Antonietta Germanà [1]

[1] Department of Agricultural, Food and Forestry Sciences (SAAF), University of Palermo, Viale delle Scienze, Ed. 4, 90128 Palermo, Italy; mariaantonietta.germana@unipa.it
[2] Department of Agricultural, Food and Environmental Sciences, University of Perugia, Borgo XX Giugno 74, 06121 Perugia, Italy; luca.regni@unipg.it
[3] Department of Agriculture and Forest Sciences, University of Tuscia, Via San Camillo de Lellis snc, 01100 Viterbo, Italy; silvestri.c@unitus.it
* Correspondence: irene.granata@unipa.it (I.G.); maurizio.micheli@unipg.it (M.M.)

**Abstract:** Salinity stress represents an increasing issue for agriculture and has a great negative impact on plant growth and crop production. The selection of genotypes able to tolerate salt stress could be a suitable solution to overcome the problem. In this context, in vitro cultures can represent a tool for identifying the NaCl tolerant genotypes and quickly producing large populations of them. The possibility of exerting selection for tolerance to NaCl by using encapsulation technology was investigated in two genotypes of fig: 'Houmairi' and 'Palazzo'. The effects of five concentrations of NaCl (0, 50, 100, 150 and 200 mM) added to the artificial endosperm were tested on the conversion of synthetic seeds and on the growth of derived shoots/plantlets. Moreover, proline (Pro) and malondialdehyde (MDA), the enzymatic activities of catalase (CAT), guaiacol peroxidase (POD), and EL (Electrolytic Leakage), as well as the chlorophyll content, flavanols, anthocyanins, and Nitrogen Balance Index (NBI) were determined on shoots/plantlet. The obtained results clearly showed that 'Houmairi' and 'Palazzo' could tolerate salt stress, although a strong difference was found depending on each specific physiological pathway. Indeed, 'Houmairi' was revealed to be more tolerant than 'Palazzo', with different response mechanisms to salt stress. The use of encapsulated vitro-derived explants proved to be a useful method to validate the selection of genotypes tolerant to salinity stress. Further investigation in the field must validate and confirm the legitimacy of the approach.

**Keywords:** salinity; abiotic stress; synthetic seed; micropropagation; fig tree; genotype selection; climate change



## 1. Introduction

Salinity represents one of the most harmful and critical factors among the abiotic stresses, limiting plants growth and crop production of numerous species of agricultural interest [1]. Recent studies [2], based on available data from the Harmonized World Soil Database, estimate that over one billion hectares of arable lands worldwide are affected by salinity and sodicity. This occurs mainly in arid and semi-arid regions where irrigation is essential to ensure satisfactory production levels, both in terms of quality and quantity, although it is carried out with poor quality or brackish irrigation water, worsening the phenomenon. Soil salinization, caused by both natural and anthropogenic activities, is estimated to increase with time at a rate of 10% [2,3]. The excess of soluble salts—most notably NaCl—in water and soil causes complex and detrimental effects on biochemical pathways of plants resulting in morphological and physiological alterations [4,5]. Salinity stress generally affects plants development in two possible ways [6]. The first one is described as ion accumulation (when the uptake of $Na^+$ and $Cl^-$ are excessively) in plant

cells, which causes an ion imbalance. The other one is the osmotic stress, which ensues by reducing the water potential, limiting the water and other soluble salts uptake by the roots due to osmotic pressure [6]. Toxic ions are accumulated in plants cells, exceeding the threshold level and causing ion toxicity [7,8]. Thus, as a result of water deficits, this leads to stomatal closure, decreasing $CO_2$ availability, and decreasing photosynthesis rate; an oxidative stress increasing the likelihood of reactive oxygen species (ROS) formation in plants also occurs. Salinity stress leads to several consequences regarding morphological traits [9], such as growth reduction—including the number and size of leaves and roots—due to the trigger of both chlorophyll content decrease and cell membrane damage caused by ion imbalance and cations outflow [10,11]. A low photosynthetic rate and high levels of osmotic stress leads to an excess of reactive oxygen species (ROS), causing cytotoxic effects and promoting specific defense pathways. The major strategies that plants use to overcome these stresses include the control of water loss through stomata closure, metabolic adjustment, toxic ion homeostasis, and osmotic adjustment [7] through the activation of enzymatic and non-enzymatic activities such as superoxide dismutase (SOD), catalase (CAT), peroxidase (POD), malondialdehyde (MDA), carotenoids, flavonoids, and the osmolyte proline, all to reduce oxidative stress and prevent further damage [12–14].

Most fruit tree species are sensitive to salinity, with only a few being considered to be moderately tolerant [4,12]. Even so, conventional methods for the selection and propagation of salinity-tolerant genotypes are both costly and time-consuming. To overcome this constraint, in vitro culture techniques represent useful tools that allow the selection and rapid validation of suitable genotypes under controlled conditions [8]. Therefore, the in vitro culture approach is helpful in investigating the impact of abiotic stress, especially salinity, in numerous Mediterranean species, owing to the significant body of literature focusing on the 0 to 400 mM range [15–17]. Several types of plant material such as cell suspension, meristem tip culture, or axillary buds can be employed for this purpose. Among the latest advancements in tissue culture applications, encapsulation is a recent technology that can be very useful in combining the benefits of in vitro culture—such as a high proliferation rate and mass production of true-to-type plants—with the easy handling and conservation of genetic resources. Additionally, this technology is now being considered as an efficient method for short-term and medium-term storage, as well as for the simplified exchange of plant material between tissue culture laboratories and for mass clonal propagation of important commercial plant species by producing synthetic seeds [18,19]. The fig tree (*Ficus carica* L.), belonging to the *Moraceae* family, is one of the most ancient and traditionally cultivated fruit trees in the Mediterranean region, grown for fresh and dried fruits. In southern Italy, the fig tree is often cultivated together with other species (i.e., olive, almond, pomegranate, apricot and grapevine). This plant can be considered moderately tolerant to salt and drought stress, even though common salt stress conditions reduce the number and length of newly formed shoots [20] and negatively affect the plant growth and fruit yield, especially in plants derived from the cuttings propagation method as they are characterized by a shallow and fibrous root system [21]. Therefore, the selection, conservation, and propagation of a wide range of salt tolerance genotypes is essential for the availability of new varieties and for their use in breeding programs, considering that the phenomenon of salinity is expected to increase with climate change over the next few years. In this context, few study have been conducted to assess genetic diversity among the Italian fig germplasm [22,23]. This study aims to improve the understanding of the physiological and morphological effects of salt stress at five different NaCl concentrations (0, 50, 100 and 200 mM) in two traditional fig genotypes (cvs. Houmairi and Palazzo) typical of the Mediterranean area, and for the first time, report on the application of encapsulation technology for synthetic seed production.

## 2. Materials and Methods

### 2.1. Plant Material

The experimental material used were derived from in vitro proliferated shoots of two *Ficus carica* L. genotypes, cvs. Houmairi and Palazzo. Subcultures were performed every 45 days in 500 mL glass jars containing 100 mL of a hormone free, full-strength MS medium [9]. In each jar, 9 binodal shoots were cultured, and the vessels were placed into a growth chamber at $23 \pm 2$ °C, under a photon flux density of $40$ μmol$^{-2}$ s$^{-1}$ with a 16/8 h light/dark photoperiod. From the proliferated shoots derived from a 45-days-old single subculture, microcuttings (uninodal segments 3–4 mm long with lateral bud) were excised and leaflets were removed and selected as the explant source.

### 2.2. Encapsulation Procedure and Culture Conditions

The microcuttings described above were initially dipped into a (2.5% *w/v*) sodium alginate (Sigma Aldrich, St. Louis, MI, USA, medium viscosity) solution for a few minutes for the coating phase, then soaked in a (1.1% *w/v*) calcium chloride ($CaCl_2 \cdot 2H_2O$) complexing solution for 35 min. Finally, the encapsulated microcuttings were rinsed with a sterile washing solution (artificial endosperm solution) for 15 min.

All three solutions were enriched with the components of artificial endosperm composed by MS basal salts, including vitamins [9] at half strength, 50 g L$^{-1}$ of sucrose, and supplemented with five different sodium chloride (NaCl) (Sigma Aldrich, St. Louis, MI, USA) concentrations (0, 50, 100, 150 and 200 mM). The pH was corrected to 5.8 with HCl and KOH 1 M solutions before autoclaving sterilization.

The hardened capsules were sown in $100 \times 15$ mm Petri dishes (five capsules in each plate) containing 10 mL of sterile culture media composed by a full strength MS medium, enriched with 30 g L$^{-1}$ of sucrose and 7 g L$^{-1}$ of agar (Plant Agar, Duchefa, BH Haarlem, Netherlands). The pH was adjusted to 5.8 before sterilization (autoclave, 20 min, 121 °C).

The obtained Petri dishes were placed in a growth chamber at $23 \pm 2$ °C, under a photon flux density of $40$ μmol$^{-2}$ s$^{-1}$ with a 16/8 h light/dark photoperiod.

All the plant material handlings were carried out in sterile conditions using a horizontal laminar flow cabinet.

### 2.3. Data Collection

The in vitro culture of both fig genotype capsules was carried out using five replicates (single Petri dish) per treatment, in addition to a 'control group' consisting of non-encapsulated propagules. In each replicate, five synseeds were sown, while NaCl concentrations consisting of 0, 50, 100, 150 and 200 mM were applied in the five treatments.

Data were collected at the end of 45 days of experimental observation, and the following parameters were evaluated: (1) Viability (% of explants with a green appearance), (2) Regrowth (% of explants that produced shoots at least 4 mm long), (3) Conversion (% of explants that produced both shoots and roots), (4) shoots and roots number per explant, (5) shoot and root length regenerated from single explants (cm).

#### 2.3.1. Measurement of Photosynthetic Pigments and Biochemical Traits

To evaluate the physiological response of the encapsulated explants to NaCl salinity, the determination of chlorophyll content, flavanols, anthocyanins, and Nitrogen Balance Index (NBI) have been carried out by using Dualex® Scientific Polyphenols and a Chlorophyll Meter (FORCE-A, Orsay, France). NBI is given as a ratio between the amounts of chlorophyll and flavonoids, the latter expressed as the ratio of chlorophyll and flavanols according to the protocols reported by Silvestri et al. (2017) and Bashir et al. (2021) [24,25]. The level of lipid peroxidation has been expressed as malondialdehyde (MDA) content, and was determined as TBA (2-thiobarbituric acid) reactive metabolites according to Astolfi et al. [26]. Briefly, fresh tissues (0.2 g) were homogenized in 10 mL of 0.25% TBA made in 10% TCA (trichloroacetic acid). The extracts were heated at 95 °C for 30 min and then quickly cooled on ice. After centrifugation at $10,000 \times g$ for 10 min, the absorbance of the

supernatant was measured at 532 nm. Correction of non-specific turbidity was made by subtracting the absorbance value taken at 600 nm. The level of lipid peroxidation was expressed as mmol $g^{-1}$ fresh weight by using an extinction coefficient of 155 mM $cm^{-1}$.

Freshly harvested leaf samples (100 mg of fresh weight) were collected, and proline concentration was determined using a spectrophotometer according to Bates et al. (1973) [27]. Briefly, for colorimetric determination, based on the proline's reaction with ninhydrin, a 1:1:1 solution of proline, ninhydrin acid, and glacial acetic acid was incubated at 100 °C for 1 h. The reaction was arrested in an iced bath and the chromophore was extracted with 4 mL toluene, and the absorbance at 520 nm was determined with a spectrophotometer EVO 60 (Thermo Fischer Scientific Inc., Waltham, MA, USA).

For ROS-scavenging enzymes, shoot tissues (1 g) were powdered in a pre-chilled mortar with liquid N2. A cold extraction buffer, containing 50 mM HEPES-KOH (pH 7.4), 5 mM $MgCl_2$, 1 mM EDTA, 10% (*v/v*) glycerol, 0.1% (*v/v*) Triton X-100, 5 mM dithiothreitol (DTT), 1 mM phenylmethylsulphonyl fluoride (PMSF), and 1% (*w/v*) polyvinylpyrrolidone (PVP) was added in a ratio of 1:7 (*w/v*) (Silvestri et al., 2021) [28].

Peroxidase activity was measured spectrophotometrically at 470 nm using guaiacol as the hydrogen donor [29]. Catalase activity (E.C. 1.11.1.6) was evaluated by measuring the decrease in absorbance at 240 nm due to the decomposition of $H_2O_2$, as described by Santangelo [29]. All reported enzyme activities were linear with time and proportional to the amount of extract used. Protein content was estimated according to Bradford (1976) [30], using BSA as the standard.

Furthermore, electrolyte leakage (EL) was used to assess cell membrane permeability (Lutts et al., 1996) [31]. Leaves were placed in 20 mL of deionized water and incubated overnight at 25 °C on a rotary shaker. The electrical conductivity of the solution ($EL_0$) was determined after 24 h. Samples were then autoclaved at 120 °C for 20 min and a last conductivity reading (EL1) were performed. The electrolyte leakage was expressed (L0/L1) as a percentage.

### 2.3.2. Statistical Analysis

The experiment was developed with a complete randomized design, with two factors studied (genotype and treatment) and their interaction.

Statistical analysis was conducted with the IBM SPSS Statistics software ((IBM Inc., New York, NY, USA; 28.0.1.1 version) using analysis of variance (two-way ANOVA), and the means were separated with Tukey's test ($p < 0.05$). Data on percentages were arcsine-transformed before performing statistical analysis [32]. The results are expressed as the mean ± standard error (SE).

## 3. Results

### 3.1. Effect of Salt Concentrations on Growth Traits

At the end of the experiment, synthetic seeds of fig genotypes 'Houmairi' and 'Palazzo' showed significant differences, cv in particular. Houmairi proved to be more tolerant than cv. Palazzo was in all the studied morphological descriptors as shown in Table 1. Both genotypes achieved satisfactory percentages, especially for viability (97.6% and 84%) and regrowth (90.4% and 65.6%); however, 'Houmairi' reached valuable performances, especially for rooting and conversion rates (66.4%), compared to 'Palazzo' (40.0 and 37.6%). The different salt concentrations, as expected, clearly affected all the examined parameters (showed in Table 1), reducing the growth rates of both genotypes, especially in the 200 mM treatments for rooting (28%) and conversion (26%), compared to the 80% of the control.

**Table 1.** Effect of five different mM concentrations of NaCl added to the artificial endosperm of synthetic seeds of *Ficus carica* L. cvs. Houmairi and Palazzo on viability, regrowth, rooting, and conversion, observed after 45 days.

| Genotype | | Viability (%) | Regrowth (%) | Rooting (%) | Conversion (%) |
|---|---|---|---|---|---|
| Houmairi | NaCl concentrations | 97.6 ± 0.17 a | 90.4 ± 0.34 a | 66.4 ± 0.41 a | 66.4 ± 0.41 a |
| Palazzo | NaCl concentrations | 84 ± 0.29 b | 65.6 ± 0.34 b | 40 ± 0.28 b | 37.6 ± 0.31 b |
| Treatment | | | | | |
| 0 (Control) | Genotypes average | 96.0 ± 0.19 | 88.0 ± 0.28 | 80.0 ± 0.3 a | 80.0 ± 0.3 a |
| 50 | Genotypes average | 96.0 ± 0.20 | 84.0 ± 0.36 | 62.0 ± 0.46 ab | 60.0 ± 0.47 ab |
| 100 | Genotypes average | 92.0 ± 0.27 | 82.0 ± 0.39 | 54.0 ± 0.34 ab | 54.0 ± 0.34 ab |
| 150 | Genotypes average | 88.0 ± 0.32 | 66.0 ± 0.36 | 42.0 ± 0.17 bc | 40.0 ± 0.19 bc |
| 200 | Genotypes average | 82.0 ± 0.27 | 70.0 ± 0.45 | 28.0 ± 0.3 c | 26.0 ± 0.34 c |
| Genotype × NaCl concentrations | NaCl (mM) | | | | |
| Houmairi | 0 (Control) | 100 ± 0.00 | 100 ± 0.00 | 92 ± 0.25 | 92 ± 0.25 |
| | 50 | 100 ± 0.00 | 100 ± 0.00 | 84 ± 0.31 | 84 ± 0.31 |
| | 100 | 100 ± 0.00 | 100 ± 0.00 | 76 ± 0.28 | 76 ± 0.28 |
| | 150 | 96 ± 0.25 | 72 ± 0.37 | 52 ± 0.19 | 52 ± 0.19 |
| | 200 | 92 ± 0.25 | 80 ± 0.37 | 28 ± 0.30 | 28 ± 0.30 |
| Palazzo | 0 (Control) | 92 ± 0.25 | 76 ± 0.10 | 68 ± 0.19 | 68 ± 0.19 |
| | 50 | 92 ± 0.25 | 68 ± 0.34 | 40 ± 0.37 | 36 ± 0.37 |
| | 100 | 84 ± 0.31 | 64 ± 0.37 | 32 ± 0.12 | 32 ± 0.12 |
| | 150 | 80 ± 0.37 | 60 ± 0.40 | 32 ± 0.12 | 28 ± 0.12 |
| | 200 | 72 ± 0.12 | 60 ± 0.48 | 28 ± 0.33 | 24 ± 0.40 |
| Significance of ANOVA | | | | | |
| Genotype | | *** | *** | *** | *** |
| NaCl concentrations | | ** | * | *** | *** |
| Genotype × NaCl concentrations | | ns | ns | ns | ns |

Mean values shown in columns according to Tukey's test at $p \leq 0.05$. Data are presented as mean ± SD. Mean values followed by the same letter are not significantly different at $p \leq 0.05$. ns, ***, **, *: non-significant or significant at $p \leq 0.001$, $p \leq 0.01$, $p \leq 0.05$, respectively.

Similarly, significant differences between the two fig varieties were observed for the other morpho-physiological traits investigated, as presented in Table 2.

Significant differences were found in the number of shoots produced during the experiment between the two studied cultivars. In cv. Houmairi in particular, every average value significantly differed from the control (0 mM), which had the higher value (1.40). Meanwhile, the mean number of shoots identified in cv. Palazzo was not significantly different from the control. Moreover, these observations were comparable to the lowest mean exhibited by 'Houmairi' (under a concentration of 200 mM).

For the shoot length parameter, cv. Houmairi showed an average length of 2.40 cm in the control; the treatment with 50 mM yielded the highest length value (3.14 cm), while subsequent treatments demonstrated decreasing values until the lowest was reached at 200 mM concentration (0.34 cm). On the contrary, cv. Palazzo showed no significant differences among treatments, as all mean values were similar to Houmairi's shorter length.

Differences in root number and length between treatments were also observed (Figures 1 and 2).

The cv. Houmairi started to show a significant decrease in the number of newly formed roots and root length at the concentration of 150 mM (1.12 and 1.17 cm) in comparison to the control (2.20 and 4.30 cm). In contrast, cv. Palazzo exhibited a significant reduction, since the lower salt concentration (50 mM) (0.60) and all the remainder salt concentrations were equivalent to the least mean values recorded for 'Houmairi' at 200 mM (0.32 and 0.57 cm), as presented in Table 2.

**Table 2.** Effect of five different concentrations of NaCl added to the artificial endosperm synthetic seeds of *Ficus carica* L. cvs. Houmairi and Palazzo on proliferation and rooting performance, observed after 45 days.

| Genotype | | Number of Shoots | Shoots Length (cm) | Number of Roots | Roots Length (cm) |
|---|---|---|---|---|---|
| Houmairi | NaCl concentrations | 0.82 ± 0.50 a | 1.60 ± 1.39 a | 0.59 ± 0.85 b | 3.06 ± 2.19 a |
| Palazzo | NaCl concentrations | 0.28 ± 0.16 b | 0.23 ± 0.19 b | 0.66 ± 0.64 a | 0.94 ± 1.35 b |
| Treatment | | | | | |
| 0 (Control) | Genotypes average | 0.92 ± 0.59 a | 1.29 ± 1.30 ab | 1.92 ± 0.58 a | 3.78 ± 1.13 a |
| 50 | Genotypes average | 0.70 ± 0.41 ab | 1.81 ± 1.77 a | 1.42 ± 1.01 ab | 2.82 ± 2.64 ab |
| 100 | Genotypes average | 0.46 ± 0.40 bc | 0.64 ± 0.62 bc | 1.24 ± 0.92 b | 2.29 ± 2.46 b |
| 150 | Genotypes average | 0.40 ± 0.36 bc | 0.58 ± 0.94 bc | 0.72 ± 0.49 c | 0.70 ± 0.56 c |
| 200 | Genotypes average | 0.26 ± 0.13 c | 0.24 ± 0.17 c | 0.32 ± 0.23 c | 0.40 ± 0.43 c |
| Genotype × NaCl concentrations | NaCl (mM) | | | | |
| Houmairi | 0 (Control) | 1.40 ± 0.40 a | 2.40 ± 0.92 ab | 2.20 ± 0.42 a | 4.30 ± 0.81 a |
| | 50 | 1.10 ± 0.30 ab | 3.14 ± 1.61 a | 2.24 ± 0.52 a | 4.96 ± 1.88 a |
| | 100 | 0.72 ± 0.41 bc | 1.14 ± 0.50 bc | 2.08 ± 0.36 a | 4.29 ± 1.90 a |
| | 150 | 0.56 ± 0.48 bc | 1.03 ± 1.23 bc | 1.12 ± 0.30 bc | 1.17 ± 0.35 bc |
| | 200 | 0.36 ± 0.09 c | 0.34 ± 0.18 c | 0.32 ± 0.18 c | 0.57 ± 0.52 c |
| Palazzo | 0 (Control) | 0.44 ± 0.22 c | 0.21 ± 0.11 c | 1.64 ± 0.62 ab | 3.28 ± 1.25 ab |
| | 50 | 0.36 ± 0.09 c | 0.49 ± 0.26 c | 0.60 ± 0.60 c | 0.67 ± 0.82 c |
| | 100 | 0.21 ± 0.14 c | 0.15 ± 0.11 c | 0.40 ± 0.14 c | 0.28 ± 0.13 c |
| | 150 | 0.24 ± 0.09 c | 0.14 ± 0.05 c | 0.32 ± 0.23 c | 0.23 ± 0.17 c |
| | 200 | 0.16 ± 0.09 c | 0.14 ± 0.10 c | 0.32 ± 0.30 c | 0.22 ± 0.26 c |
| **Significance of ANOVA** | | | | | |
| Genotype | | *** | *** | *** | *** |
| NaCl concentrations | | *** | *** | *** | *** |
| Genotype × NaCl concentrations | | ** | *** | *** | *** |

Mean values shown in columns according to Tukey's test at $p \leq 0.05$. Data are presented as mean ± SD. Mean values followed by the same letter are not significantly different at $p \leq 0.05$. ***, **: significant at $p \leq 0.001$, $p \leq 0.01$, respectively.

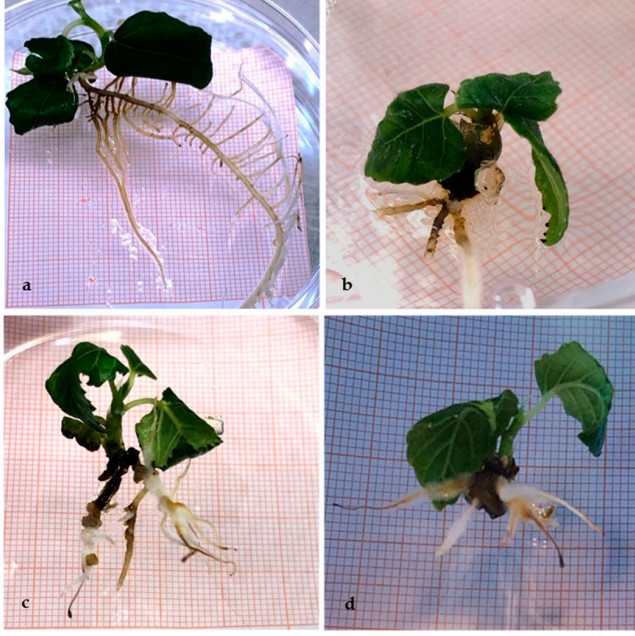

**Figure 1.** (**a**) 'Houmairi' control; (**b**) 'Palazzo control; (**c**) 'Houmairi' 50 mM; (**d**) 'Palazzo' 50 mM, at the end of the experiment.

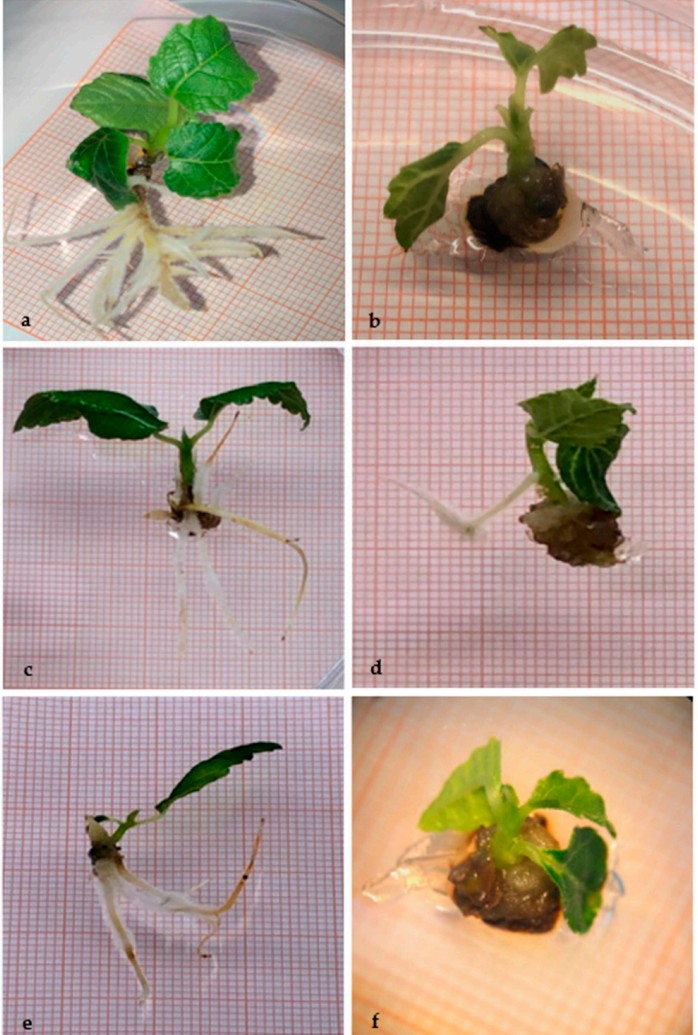

**Figure 2.** (**a**) 'Houmairi' 100 mM; (**b**) 'Palazzo' 100 mM; (**c**) 'Houmairi' 150 mM; (**d**) 'Palazzo' 150 mM; (**e**) 'Houmairi' 200 mM; (**f**) 'Palazzo' 200 mM at the end of the experiment.

### 3.2. Effect of Alginate Coating Treatment on Growth Activity

The synthetic seeds exposed at 0 mM of NaCl (control) were also compared to the propagules lacking the encapsulation matrix; at the end of the experiments, growth performance was evaluated. In Table 3, percentages of viability, regrowth, rooting, and conversion are detailed.

No statistical differences have been observed in the viability and regrowth rates between the varieties 'Houmairi' and 'Palazzo'. On the other hand, there were differences in rooting and conversion rates between the cultivars. Indeed, 'Houmairi' yielded more rooted plantlets compared with the cv. Palazzo (84 and 58%). Additionally, providing the coating treatment resulted in the highest mean values for both genotypes when compared to non-coated propagules. The number of roots and their length prove to be closely related to genotype, as illustrated in Table 3. 'Houmairi' showed higher average values (2.36 and 4.33 cm) according to the other morphological traits previously described.

Likewise, the number and length of shoots were positively influenced by the alginate coating treatment, especially in 'Houmairi' genotype (0.94 and 1.46 cm) as shown in Table 4.

**Table 3.** Effect of coating treatment on *Ficus carica* L. cvs. Houmairi and Palazzo microcuttings on viability, regrowth, rooting and conversion, and rooting performance observed after 45 days.

| Genotype | | Viability (%) | Regrowth (%) | Rooting (%) | Conversion (%) | Number of Roots | Roots Length (cm) |
|---|---|---|---|---|---|---|---|
| Houmairi | Treatments average | 100 ± 0.00 | 94 ± 0.13 | 84 ± 0.21 a | 84 ± 0.21 a | 2.36 ± 0.69 a | 4.33 ± 1.48 a |
| Palazzo | Treatments average | 96 ± 0.20 | 82 ± 0.15 | 58 ± 0.20 b | 58 ± 0.20 b | 1.26 ± 0.67 b | 2.32 ± 1.42 b |
| Treatment | | | | | | | |
| No coating | Genotypes average | 100 ± 0.00 | 88 ± 0.30 | 62 ± 0.4 b | 62 ± 0.4 b | 1.70 ± 1.1 | 2.90 ± 2.2 |
| Coating | Genotypes average | 96 ± 0.20 | 88 ± 0.30 | 80 ± 0.3 a | 80 ± 0.3 a | 1.90 ± 0.6 | 3.80 ± 1.1 |
| Genotype × Treatment | | | | | | | |
| Houmairi | No coating | 100 ± 0.00 | 88 ± 0.18 | 76 ± 0.26 | 76 ± 0.26 | 2.52 ± 0.91 | 4.36 ± 2.06 |
| | Coating | 100 ± 0.00 | 100 ± 0.00 | 92 ± 0.11 | 92 ± 0.11 | 2.21 ± 0.42 | 4.30 ± 0.81 |
| Palazzo | No coating | 100 ± 0.00 | 88 ± 0.18 | 48 ± 0.18 | 48 ± 0.18 | 0.88 ± 0.52 | 1.37 ± 0.85 |
| | Coating | 92 ± 0.11 | 76 ± 0.09 | 68 ± 0.18 | 68 ± 0.18 | 1.64 ± 0.62 | 3.27 ± 1.25 |
| Significance of ANOVA | | | | | | | |
| Genotype | | ns | ns | ** | ** | * | * |
| Treatment | | ns | ns | * | * | ns | ns |
| Genotype × Treatment | | ns | ns | ns | ns | ns | ns |

Mean values shown in columns according to Tukey's test at $p \leq 0.05$. Data are presented as mean ± SD. Mean values followed by the same letter are not significantly different, at $p \leq 0.05$. ns, **, *: non-significant or significant at $p \leq 0.01$, $p \leq 0.05$, respectively.

**Table 4.** Effect of coating treatment on *Ficus carica* L. cvs. Houmairi and Palazzo microcuttings on proliferation activities observed after 45 days.

| Genotype | Treatment | Number of Shoots | Shoots Length (cm) |
|---|---|---|---|
| Houmairi | Treatments average | 0.94 ± 0.61 a | 1.46 ± 1.23 a |
| Palazzo | Treatments average | 0.44 ± 0.16 b | 0.20 ± 0.08 b |
| Treatment | | | |
| No coating | Genotypes average | 0.5 ± 0.3 b | 0.4 ± 0.50 b |
| Coating | Genotypes average | 0.9 ± 0.6 a | 1.3 ± 1.03 a |
| Genotype × Treatment | | | |
| Houmairi | No coating | 0.48 ± 0.39 b | 0.56 ± 0.70 b |
| | Coating | 1.40 ± 0.40 a | 2.37 ± 0.92 a |
| Palazzo | No coating | 0.44 ± 0.09 b | 0.19 ± 0.05 b |
| | Coating | 0.44 ± 0.22 b | 0.21 ± 0.11 b |
| Significance of ANOVA | | | |
| Genotype | | *** | *** |
| Treatment | | *** | *** |
| Genotype × Treatment | | ** | *** |

Mean values shown in columns according to Tukey's test at $p \leq 0.05$. Data are presented as mean ± SD. Mean values followed by the same letter are not significantly different at $p \leq 0.05$. ***, **: significant at $p \leq 0.001$, $p \leq 0.01$, respectively.

### 3.3. Variability on Photosynthetic Pigments and Biochemical Traits

The results of the analysis of variance suggest that the increasing salt concentrations had a significant effect on all the parameters studied (Table 5). Additionally, regarding the chlorophyll content, no significative differences have been found between the cv. Houmairi and Palazzo, but an increase in salt concentration at 200 mM induce an increase in chlorophyll content (30.11 and 30.97 µg/cm$^2$).

Flavanols and anthocyanins, two important traits related to the secondary metabolism of the shoots, showed clear differences due to the cultivars, but also due to the salt concentration used. Furthermore, a significant interaction between factors (cv × treatments) were observed. In control conditions (0 mM NaCl), flavanols contents were not different between the cv. Houmairi and Palazzo (0.42 and 0.39 µg/cm$^2$, respectively).

As the concentration of NaCl increased, the cultivars showed different behaviors: in cv. Houmairi, in fact, no significant difference was observed between the control and increasing concentrations of NaCl, while in the cv. Palazzo at the NaCl concentration of 100 mM, a significant increase in this parameter was observed compared to the control

(0.54 vs. 0.39 µg/cm$^2$). At higher NaCl concentrations (150 and 200 mM), the average flavanol content was lower than the treatment with 100 mM NaCl.

However, regarding the anthocyanin content, different behavior of the two cultivars was observed when increasing the salt treatments (as also confirmed by the significance of the interaction between the two factors). In cv. Houmairi, in fact, at the salt concentration of 100 mM, a significant increase in anthocyanin content was observed. In cv. Palazzo, on the other hand, a significant decrease in anthocyanin content was observed in the presence of different salt concentrations.

As the NBI parameter is an index that takes into account the relationship between the primary and secondary metabolism, it depends on the content of chlorophylls, anthocyanins, and flavanols. In this case, however, apart from a statistical difference between the cultivars, this parameter showed no difference among the treatments.

**Table 5.** Estimates of physiological parameters detected through the Dualex® tool on fig genotypes 'Houmairi' and 'Palazzo' exposed to five different salt (NaCl) concentrations.

| Genotype | | Chlorophyll (µg/cm$^2$) | Flavanols (µg/cm$^2$) | Anthocyanin (µg/cm$^2$) | NBI (Chl./Flav.) |
|---|---|---|---|---|---|
| Houmairi | NaCl concentrations | 28.65 ± 3.83 | 0.42 ± 0.06 b | 0.07 ± 0.05 | 70.27 ± 15.76 a |
| Palazzo | NaCl concentrations | 27.60 ± 4.90 | 0.46 ± 0.09 a | 0.07 ± 0.03 | 62.56 ± 14.48 b |
| Treatment | | | | | |
| 0 (Control) | Genotypes average | 25.63 ± 4.57 b | 0.41 ± 0.05 c | 0.08 ± 0.04 ab | 63.57 ± 12.67 |
| 50 | Genotypes average | 29.00 ± 4.00 ab | 0.44 ± 0.06 ab | 0.06 ± 0.03 bc | 66.90 ± 15.60 |
| 100 | Genotypes average | 28.50 ± 3.68 ab | 0.48 ± 0.11 a | 0.05 ± 0.03 c | 62.68 ± 17.25 |
| 150 | Genotypes average | 27.63 ± 4.87 ab | 0.44 ± 0.07 ab | 0.10 ± 0.03 a | 64.75 ± 17.29 |
| 200 | Genotypes average | 30.11 ± 3.88 a | 0.41 ± 0.06 c | 0.07 ± 0.04 abc | 74.17 ± 13.03 |
| Genotype × NaCl concentrations | NaCl (mM) | | | | |
| Houmairi | 0 (Control) | 26.05 ± 6.95 | 0.42 ± 0.05 b | 0.07 ± 0.04 bc | 62.63 ± 22.69 |
| | 50 | 28.69 ± 5.83 | 0.45 ± 0.06 ab | 0.06 ± 0.04 bc | 65.81 ± 20.78 |
| | 100 | 28.89 ± 4.07 | 0.42 ± 0.06 b | 0.04 ± 0.02 c | 70.82 ± 17.86 |
| | 150 | 30.38 ± 3.67 | 0.42 ± 0.09 b | 0.13 ± 0.02 a | 74.93 ± 15.25 |
| | 200 | 29.25 ± 8.59 | 0.39 ± 0.06 b | 0.07 ± 0.05 bc | 77.14 ± 26.75 |
| Palazzo | 0 (Control) | 25.22 ± 5.35 | 0.39 ± 0.06 b | 0.10 ± 0.04 ab | 64.52 ± 5.86 |
| | 50 | 29.31 ± 3.96 | 0.44 ± 0.06 b | 0.06 ± 0.02 bc | 68.00 ± 9.24 |
| | 100 | 28.12 ± 4.57 | 0.54 ± 0.12 a | 0.07 ± 0.03 bc | 54.54 ± 11.25 |
| | 150 | 24.89 ± 4.53 | 0.46 ± 0.05 ab | 0.07 ± 0.02 bc | 54.56 ± 8.41 |
| | 200 | 30.97 ± 3.81 | 0.44 ± 0.05 b | 0.07 ± 0.02 bc | 71.19 ± 5.71 |
| Significance of ANOVA | | | | | |
| Genotype | | ns | ** | ns | ** |
| Treatment | | * | ** | ** | ns |
| Genotype × NaCl concentrations | | ns | ** | ** | ns |

Mean values shown in columns according to Tukey's test at $p \leq 0.05$. Data are presented as mean ± SD. Mean values followed by the same letter are not significantly different at $p \leq 0.05$. ns, **, *: non-significant or significant at $p \leq 0.01$, $p \leq 0.05$, respectively.

Many biochemical signals are generally produced in plant tissues due saline stress, and Table 6 shows the data of the major involved traits.

The electrolyte leakage (EL), an early indicator of cell membrane damage, showed different trends depending on the cultivar: in the case of cv. Houmairi, in fact, no significant differences were observed among treatments, unlike in the case of cv. Palazzo, where when salt concentration increased, particularly at NaCl concentrations of 100, 150, and 200 mM, a significant increase in this trait was observed, confirming higher cellular damage in 'Palazzo' than the 'Houmairi'.

**Table 6.** Electrolyte leakage (EL), total protein (PROT), malondialdehyde (MDA), guaiacol peroxidase (POD), catalase (CAT), and proline contents detected in fig genotypes shoots 'Houmairi' and 'Palazzo' exposed to five different salt (NaCl) concentrations.

| Genotype | | EL (%) | PROT (mg g$^{-1}$ FW) | MDA (µg g$^{-1}$ FW) | POD ($\Delta$470 min$^{-1}$·mg$^{-1}$ prot) | CAT ($\Delta$240 min$^{-1}$·mg$^{-1}$ prot) | Proline (µg g$^{-1}$ FW) |
|---|---|---|---|---|---|---|---|
| Houmairi | NaCl concentrations | 18.41 ± 3.16 b | 5.74 ± 0.91 | 4.36 ± 0.85 b | 7.25 ± 3.21 | 2.12 ± 0.23 | 3.57 ± 0.79 |
| Palazzo | NaCl concentrations | 33.25 ± 5.41 a | 5.67 ± 1.52 | 4.60 ± 1.21 a | 8.08 ± 4.05 | 2.25 ± 0.23 | 4.01 ± 1.78 |
| Treatment | | | | | | | |
| 0 (Control) | Genotypes average | 24.26 ± 5.82 | 4.98 ± 1.06 d | 4.24 ± 0.90 c | 4.21 ± 0.35 b | 2.07 ± 0.23 | 2.59 ± 0.72 b |
| 50 | Genotypes average | 23.74 ± 6.21 | 4.86 ± 0.26 d | 3.67 ± 0.06 d | 5.03 ± 0.63 b | 2.22 ± 0.23 | 3.27 ± 0.84 b |
| 100 | Genotypes average | 28.52 ± 10.31 | 6.89 ± 1.55 a | 3.77 ± 0.69 d | 6.45 ± 2.15 b | 2.31 ± 0.20 | 3.18 ± 0.54 b |
| 150 | Genotypes average | 25.62 ± 11.72 | 6.56 ± 0.84 b | 5.72 ± 1.10 a | 10.81 ± 3.40 a | 2.11 ± 0.26 | 4.73 ± 1.17 a |
| 200 | Genotypes average | 27.01 ± 10.28 | 5.22 ± 0.39 c | 5.01 ± 0.26 b | 11.83 ± 1.40 a | 2.23 ± 0.25 | 5.18 ± 1.47 a |
| Genotype × NaCl concentrations | NaCl (mM) | | | | | | |
| Houmairi | 0 (Control) | 20.07 ± 3.14 bcd | 5.15 ± 0.12 b | 5.05 ± 0.10 b | 4.14 ± 0.24 d | 2.16 ± 0.30 | 2.87 ± 0.96 c |
| | 50 | 19.19 ± 3.85 bcd | 5.06 ± 0.21 b | 3.63 ± 0.05 d | 4.68 ± 0.44 cd | 2.07 ± 0.14 | 3.15 ± 0.30 bc |
| | 100 | 19.37 ± 0.63 bcd | 5.49 ± 0.16 c | 3.14 ± 0.08 e | 7.5 ± 2.56 bc | 2.21 ± 0.23 | 3.26 ± 0.82 c |
| | 150 | 15.49 ± 3.03 d | 7.32 ± 0.12 a | 4.72 ± 0.05 c | 7.98 ± 1.93 b | 1.93 ± 0.05 | 3.84 ± 0.74 bc |
| | 200 | 17.92 ± 4.01 cd | 4.88 ± 0.12 c | 5.24 ± 0.04 a | 11.94 ± 1.85 a | 2.22 ± 0.34 | 3.93 ± 0.83 bc |
| Palazzo | 0 (Control) | 28.45 ± 4.71 ab | 4.01 ± 0.04 e | 3.42 ± 0.06 e | 4.28 ± 0.49 c | 1.98 ± 0.13 | 2.30 ± 0.35 c |
| | 50 | 28.28 ± 4.42 abc | 4.66 ± 0.07 d | 3.70 ± 0.05 d | 5.38 ± 0.65 c | 2.36 ± 0.23 | 2.60 ± 0.58 c |
| | 100 | 37.68 ± 3.74 a | 8.30 ± 0.12 a | 4.40 ± 0.10 c | 5.38 ± 1.27 c | 2.40 ± 0.12 | 3.11 ± 0.17 c |
| | 150 | 35.75 ± 5.11 a | 5.80 ± 0.11 b | 6.72 ± 0.08 a | 13.64 ± 1.05 a | 2.30 ± 0.25 | 5.63 ± 0.68 ab |
| | 200 | 36.10 ± 0.60 a | 5.57 ± 0.12 c | 4.77 ± 0.12 b | 11.72 ± 1.20 b | 2.23 ± 0.21 | 6.42 ± 0.16 a |
| Significance of ANOVA | | | | | | | |
| Genotype | | ** | ns | ** | ns | ns | ns |
| Treatment | | ns | ** | ** | ** | ns | ** |
| Genotype × NaCl concentrations | | * | ** | ** | ** | ns | ** |

Mean values shown in columns according to Tukey's test at $p \leq 0.05$. Data are presented as mean ± SD. Mean values followed by the same letter are not significantly different at $p \leq 0.05$. ns, **, *: non-significant or significant at $p \leq 0.01$, $p \leq 0.05$, respectively.

The total protein content trend—a crucial estimate that provides information on the amount of stress-related proteins—followed a similar pattern for both genotypes, but 'Palazzo' showed a spike in the 100 mM salt treatment, while 'Houmairi' showed an increase in 150 mM treatment.

Also, the content of malondialdehyde could be useful to evaluate the oxidative stress as a result of lipid peroxidation. In this case, 'Houmairi' seems to have not been affected by salt stress because at 50–100 mM of NaCl, it showed lower mean values than the control, and only in the higher salt treatments (200 mM) was an important increase of MDA content observed compared to the control (11.94 and 4.14 respectively). At the same time, cv. Palazzo showed a significative increase of MDA content, also at a concentration of 150 mM.

A central role for ROS detoxifying signaling pathways, especially for $H_2O_2$, is the upregulation of the two groups of enzymes; i.e., catalases and peroxidases. In our findings, both POD and CAT activities in both cv showed no differences in control condition, but when subjected to salt stress, their responses were different. Regarding the POD activity, both varieties showed a significant increase in the activity of this enzyme as salt concentration increased, particularly at the 150 and 200 mM concentrations. On the other hand, neither cultivar showed any differences regarding CAT activities, not among them nor or as a function of imposed salt stress.

One of the adaptive responses to prevent salt toxicity is the osmotic adjustment through the synthesis and accumulation of low-molecular-weight metabolites known as osmolytes, especially free proline (PRO), a proteinogenic amino acid that plays a crucial role. In our experiments, the cultivar Houmairi did not show significant differences, maintaining a constant threshold in all treatments observed. The cv. Palazzo, contrarily, showed a

significant increase in proline contents when increasing the concentration of NaCl, in particular at higher concentrations of 150 and 200 mM.

## 4. Discussion

The obtained findings clearly show that fig genotypes 'Houmairi' and 'Palazzo' are affected differently by salinity stress. Both genotypes showed a medium-high level of viability even under the most severe concentrations; however, 'Houmairi' proved to be more tolerant than 'Palazzo' to salt stress in almost all the morphological indicators evaluated. The number of shoots and their length were negatively affected by the increasing salt concentrations, according to other similar studies [9,33]. However, the number of roots and their length showed a different trend; therefore, cv. Houmairi starts to be affected by salt stress from the third level equal to 150 mM. This contrasts with the findings of Vijayan et al. (2003) [34], but is in agreement with another study [35] that suggests a different Na ion accumulation in shoots and roots. Our findings are also confirmed by the assessment of polyphenolic compounds conducted by the Dualex; i.e., the chlorophyll content of 'Houmairi' shoots is still the same in all the applied treatments while 'Palazzo' shoots indicate a significantly lower value under 150 mM of salt concentration. It is fully demonstrated that a high concentration of NaCl affects the photosynthesis rate [36]. Many authors agree with our results [9,37,38]; Ekinci et al. (2012) [39] suggest that an increase in the chlorophyll content under salt stress conditions could be due to the increased number of chloroplasts and to the accumulation of NaCl in the chloroplast.

The nitrogen balance index (NBI), which is expressed as the ratio of chlorophyll to epidermal flavonoids belonging to epidermal polyphenolics (Chl/Flav), represents a sensitive indicator of plant nitrogen N status [40], because when plants are subjected to a low availability of nitrogen, the content of (Chl) decreases while epidermal polyphenolics (Phen) increase; and so, it represents an useful method for estimating stress-related metabolism parameters. In this study, both flavanols content and chlorophyll content were not significantly different in 'Houmairi' in all the treatments; cv. Palazzo, on the contrary, shows a different trend; flavanols content indeed had a growing value compared to both the control and cv. Houmairi. This evidence is almost the same for the electrolyte leakage.

When cells are critically damaged and lose the integrity of the membrane, electrolytes—K ions in particular—leak out of the cell, and it is possible to measure the electrolytic conductivity of the water in order to assess the relative of dead cells in response to stresses [41]. Salt stress induces an increase in the concentration of $Na^+$ inside the cells, and this results in an excess of $Na^+$ in the cytosol. At the same time, it induces a $K^+$ efflux, thus allowing the drastic decrease of the cytosolic $K^+/Na^+$ ratio [42]. The EL content of the 'Houmairi' genotype was not significant for all treatments, while 'Palazzo' shows greater values than 'Houmairi' and the control. Electrolyte leakage may be induced by a wide range of factors [43] in many stress-tolerant species in order to avoid stress-induced loss of $K^+$. Higher levels of $K^+$ are maintained in stress conditions [44]; this can be achieved through increased $K^+$ selectivity of total plasmatic membrane conductance [43]. Demidchik (2014) [43] suggests that the decrease in cytosolic $K^+$ could play the role of a 'switch' that reduces the energy consumption for anabolic reactions and stimulates 'energy-releasing' for catabolic processes. So, this mechanism stops plant growth and 'redirects' the energy flow to adaptation and reparation needs. This represents an interesting hypothesis and could be a critical step in plant cell adaptation to any stress factor, as plants which are subjected to stress stop growing and use the released energy to fight stress-induced injuries [43]. Different studies [45,46] reported that salinity-tolerant plants limit the excess of soluble salts in vacuoles, or accumulate essential ions in different tissues. The study of Yassin et al., 2019) [47] suggests that the low uptake of $Na^+$ and high uptake of $K^+$ denote salinity tolerance in higher plants. Probably, 'Houmairi' and 'Palazzo' could have different mechanisms to avoid salt stress.

Generally, salinity stress causes a significant increase in protein content in plant tissues, and their increased production is particularly important to stabilize cell membranes for cell survival, especially under salt stress conditions.

Salt tolerant cultivars have higher protein concentration due to their higher osmotic regulation mechanism efficiency, which in turn causes a decrease of sodium toxicity in the cytoplasm compared to susceptible ones. The result is the prevention of proteins reduction under salt stress [48]. In the present study, both cultivars—Houmairi and Palazzo—showed a significant amount of total protein content.

Lipid peroxidation (MDA) is an oxidative salt-stress related indicator; indeed, higher concentrations of MDA are related to cell membrane damages. In salt-tolerant plants, there is lower MDA expression, as well as lower ROS production—especially $H_2O_2$—due to the presence of an efficient protection mechanism and a high scavenging capacity [6]. In our study, we found that 'Palazzo' had a significant increase of MDA, especially in the range 100–200 mM, but the highest value was registered at a 150 mM level. This finding, according to [9], is probably due to the fact that when NaCl concentration was equal to 200 mM, some stress-related genes were induced in leaves and the MDA accumulation was lower. On the other hand, 'Houmairi' showed an opposite trend, since in all the treatments, with the exception of 150–200 mM, the content of MDA was lower than the control. A recent study [36] suggests that a sustained decrease of MDA expression might be caused by the activation of PSII (photosystem II) core proteins (which are responsible for water splitting, oxygen evolution, and plastoquinone reduction of the enzyme RuBisCo) [49].

Plants control the concentrations of ROS under no stressful conditions using an array of constitutive enzymatic and non-enzymatic antioxidants such as catalase (CAT) and guaicol peroxidase (POD). Generally high antioxidant activities are interpreted as symptoms of oxidative stress (the plant upregulates the antioxidant enzymes as a result of the increased production of ROS), but the higher antioxidant activity could also be interpreted as a tolerance mechanism to oxidative stress, since these plants avoid oxidative stress by maintaining higher antioxidant activity [6]. In our study, we found a higher increase of POD with the 50 mM treatment, associated with a non-significant accumulation of CAT for both genotypes 'Houmairi' and 'Palazzo'. This result is in accordance with [6,50].

Salinity occurs with excess ions in the root zone that causes a detrimental effect on water uptake, resulting in osmotic stress and high concentrations of toxic ions inside the cells. Plants have developed complex strategies to optimize adaptive responses to reduce salt toxicity. One of these mechanisms is osmotic adjustment through the synthesis and accumulation of low-molecular-weight metabolites known as osmolytes, in order to maintain metabolic activities [51] such as soluble sugars, proteins, and GSH. The large availability of these compounds in the cells could be helpful in the selection for salt tolerance genotypes [52]. Proline is a multifunctional amino acid that accumulates in plant cells subjected to several kinds of stress [51]. It may adjust ion balance, scavenge free radicals, and can act as a non-enzymatic antioxidant [53]. Many studies [49,53,54] reported that proline content increases with increasing salt stress levels. In the present study, 'Houmairi' showed a slow increase, but resulted in substantial differences among all salt treatments, while 'Palazzo' had an increase from the first salinity concentration (50 mM), and in particular, shoots exposed to 150 and 200 mM had higher values.

## 5. Conclusions

Salinity is a major constraint that negatively affects soil fertility and crop production. The aim of this study was to assess the tolerance of fig nodal explants to salt stress when encapsulated in a nutrient matrix containing varying concentrations of NaCl. The study evaluated both vegetative and rhizogenic activity of the synthetic seeds, as well as chemical factors to better understand the effects on the two varieties studied ('Houmairi' and 'Palazzo'). The results indicate that both genotypes can tolerate salt stress, but through different physiological pathways. 'Houmairi' was found to be more tolerant than 'Palazzo', and both varieties showed variability, possibly due to genetic differences. Synthetic seed

technology and tissue cultures proved to be a useful method for validating the selection of valuable genotypes.

**Author Contributions:** M.A.G., C.S. and I.G. are responsible for the conceptualization and experimental design; M.M. and C.S. for methodology, I.G., C.S. and L.R. conducted the experiment and data analysis, I.G, C.S. and M.M. draft the manuscript, M.M and L.R acquire publication costs. All authors have read and agreed to the published version of the manuscript.

**Funding:** This research received no external funding.

**Data Availability Statement:** The data presented in this study are available on request from the corresponding author. The data are not publicly available due to no public repository for raw data is provided.

**Conflicts of Interest:** The authors declare no conflict of interest.

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
