# Peer review of "Application of Encapsulation Technology: In Vitro Screening of Two Ficus carica L. Genotypes under Different NaCl Concentrations"

_horticulturae, doi:10.3390/horticulturae9121344_

Round 1
Reviewer 1 Report
Comments and Suggestions for Authors
The article "Application of the encapsulation technology: in vitro screening of two Ficus carica L. genotypes under different NaCl concentrations" lacks novelty, but the manuscript can be improved.
1) The abstract is poorly constructed and lacks ingredients. No results were mentioned, without explaining the objective and methodology. I presume this is an original research article, not communication. Overall abstract should be formulated to make it readable.
2) The introduction lacks a review of the literature and previous gaps in the study. No objective was explained in this section.
3) The methodology is confusing. Why did the authors select these concentrations of salinity? The studied attributes what are the methods? Detailed methodology for the antioxidative enzymes must be added, what is the quantity of samples, extraction was done using which media, and what standards were used to calculate the final value?
4) Results are nicely formulated but the discussion is fragmented. There is no link between the studied attributes and your results. Please explain the studied attributes and your objective.
Comments on the Quality of English LanguageMinor changes required.
Author Response
Author's Reply to the Review Report (Reviewer 1)
Comments and Suggestions for Authors
The article "Application of the encapsulation technology: in vitro screening of two Ficus carica L. genotypes under different NaCl concentrations" lacks novelty, but the manuscript can be improved.
Reply: The authors thank the reviewer for his constructive assessment. We have implemented his recommendations to improve the manuscript.
______________________________________________________________________________________
Comment 1: The abstract is poorly constructed and lacks ingredients. No results were mentioned, without explaining the objective and methodology. I presume this is an original research article, not communication. Overall abstract should be formulated to make it readable.
Reply 1: The abstract was completed of the missing parts as suggested by the reviewer. The reviewer well understood the originality of the research and by following his suggestions the contents of the paper should now be more understandable.
______________________________________________________________________________________
Comment 2: The introduction lacks a review of the literature and previous gaps in the study. No objective was explained in this section.
Reply 2: After considering the reviewer's comments, we have revised the introduction to enhance its clarity and coherence. Several other appropriate references have been added to support the argument and the objective of the study has been widely described.
______________________________________________________________________________________
Comment 3: The methodology is confusing. Why did the authors select these concentrations of salinity? The studied attributes what are the methods? Detailed methodology for the antioxidative enzymes must be added, what is the quantity of samples, extraction was done using which media, and what standards were used to calculate the final value?
Reply 3: The authors followed the instructions provided by the reviewer. The salinity levels used in the experiment were chosen based on previous experience and other studies, which are cited in the text. Furthermore, the section on antioxidant enzymes was revised to include missing information, following the suggestions provided in the bibliography for evaluating research data on these aspects.
______________________________________________________________________________________
Comment 4: Results are nicely formulated but the discussion is fragmented. There is no link between the studied attributes and your results. Please explain the studied attributes and your objective.
Reply 4: The commentary on the results was revised to improve interpretation, and the tables and statistical analysis were also enhanced based on reviewer feedback.
Reviewer 2 Report
Comments and Suggestions for Authors
The manuscrito: Application of the encapsulation technology: in vitro screening 2 of two Ficus carica L. genotypes under different NaCl concentrations, It is very relevant and of great applicability, especially when we consider global climate change, which will certainly cause enormous damage to agricultural productivity.
The introduction to the manuscript is well structured and very informative. However, the statistical design, statistical analysis and presentation of the data is very confusing, which makes it difficult to follow and trust the results.
The way the authors presented the results, I cannot evaluate the results, discussion and conclusion.
Materiais e métodos:
Line 95 ….. : What's the scheme? factorial? subdivided plots???? if there are two factors G and T and it shows the ANOVA with interaction, the table should have upper and lower case letters (row and column)
what are the factors?: 1- cultivate 2- [ ] NaCl = 2x6 factorial?
Line: 105-108: according to which methodologies????
The authors did not follow the pattern of presenting results based on statistics, since they presented means of genotypes where there was no effect of the genotypes, and they did not compare means of the interaction where there was an effect of the interaction, that is, there should have been purchases in the line and in the column, uppercase and lowercase letters.
Therefore, I am unable to evaluate the results and discussion of this manuscript.
Results:
You need to inform why you combined all treatments when there was a significant interaction between Genotype x treatment and made a single comparison of means in the column....
Table 1:
What's the scheme? factorial? subdivided plots???? if there are two factors G and T and it shows the ANOVA with interaction, the table should have upper and lower case letters (row and column)
The authors did not follow the pattern of presenting results based on statistics, since they presented means of genotypes where there was no effect of the genotypes, and they did not compare means of the interaction where there was an effect of the interaction, that is, there should have been purchases in the line and in the column, uppercase and lowercase letters.
Table 2:
If there was an interaction between the G x T factors, there should be a comparison between lines
(2 Genotypes) and columns (5 NaCl treatments), so upper and lower case letters
Table 3:
Viability and growth should only have an average value........ check the other characteristics according to the statistics
Table 4:
If there was a significant difference for the G xT interaction, there should be a comparison between line and column
Conclusion
Conclusion began with Introduction and objectives. I suggest deleting
Author Response
Author's Reply to the Review Report (Reviewer 2)
Comments and Suggestions for Authors
Comment 1
The manuscrito: Application of the encapsulation technology: in vitro screening 2 of two Ficus carica L. genotypes under different NaCl concentrations, It is very relevant and of great applicability, especially when we consider global climate change, which will certainly cause enormous damage to agricultural productivity.
Reply 1: The authors express gratitude to the reviewer for recognizing the significance of the topic proposed and investigated in the study.
______________________________________________________________________________________
Comment 2
The introduction to the manuscript is well structured and very informative. However, the statistical design, statistical analysis and presentation of the data are very confusing, which makes it difficult to follow and trust the results. The way the authors presented the results, I cannot evaluate the results, discussion and conclusion.
Reply 2: We took the reviewer's advice and concentrated on enhancing the way we presented our results and statistical analysis. This helped us to better explain our approach to data evaluation and ensure that the results were more significant.
______________________________________________________________________________________
Comment 3
Materiais e métodos:
Line 95 ….. : What's the scheme? factorial? subdivided plots???? if there are two factors G and T and it shows the ANOVA with interaction, the table should have upper and lower case letters (row and column)
what are the factors?: 1- cultivate 2- [ ] NaCl = 2x6 factorial?
Line: 105-108: according to which methodologies????
The authors did not follow the pattern of presenting results based on statistics, since they presented means of genotypes where there was no effect of the genotypes, and they did not compare means of the interaction where there was an effect of the interaction, that is, there should have been purchases in the line and in the column, uppercase and lowercase letters.
Therefore, I am unable to evaluate the results and discussion of this manuscript.
Reply 3: The authors carefully considered the feedback provided by the reviewer regarding the presentation of data and made necessary changes to the Materials and Methods section to improve its clarity and comprehensiveness. They have separated the descriptions of the data collection method and statistical analysis into distinct paragraphs, making it easier for readers to understand and follow. Additionally, they have restructured the tables according to the reviewers' suggestions in an attempt to enhance the presentation of the data. The updated version of the manuscript now reflects these changes.
______________________________________________________________________________________
Comment 4
Results: You need to inform why you combined all treatments when there was a significant interaction between Genotype x treatment and made a single comparison of means in the column....
Table 1: What's the scheme? factorial? subdivided plots???? if there are two factors G and T and it shows the ANOVA with interaction, the table should have upper and lower case letters (row and column)
The authors did not follow the pattern of presenting results based on statistics, since they presented means of genotypes where there was no effect of the genotypes, and they did not compare means of the interaction where there was an effect of the interaction, that is, there should have been purchases in the line and in the column, uppercase and lowercase letters.
Table 2: If there was an interaction between the G x T factors, there should be a comparison between lines (2 Genotypes) and columns (5 NaCl treatments), so upper and lower case letters
Table 3: Viability and growth should only have an average value........ check the other characteristics according to the statistics
Table 4: If there was a significant difference for the G xT interaction, there should be a comparison between line and column
Reply 4: The authors took the reviewer's correct observations into account when commenting on the results and organizing the tables. To make the data more readable and effective, the tables were fully revised and restructured based on the reviewer's suggestions. This was done to facilitate better comparison of the data in cases where an interaction effect is evident.
______________________________________________________________________________________
Comment 5
Conclusion
Conclusion began with Introduction and objectives. I suggest deleting
Reply 5: The authors have modified and expanded the introduction. As a result, they feel it is appropriate to include the conclusion section, as suggested by the editor. Although the conclusion is extremely concise, it is necessary to summarise what is extensively expressed in the paragraphs of the presentation of results and subsequent discussion.
Reviewer 3 Report
Comments and Suggestions for Authors
Dear Authors
The manuscript horticulturae-2675561 titled “Application of the encapsulation technology: in vitro screening of two Ficus carica L. genotypes under different NaCl concentrations” has done a vitro work to determine the different tolerance of two Fig tree cultivars. It was interesting, however, I still have some major and minor comments as follow.
Major comments
1. Authors didn’t carefully arrange the data in tables and did many mistakes in tables as I mentioned in the minor comments, please carefully rearrange them and double check them.
2. I suggest authors clearly check the results section and check all the description in English by native English speaker.
3. In line 342-343 and line 442, it is difficult to conclude that “Houmairi” in more tolerant than “Palazzo”, because the decreased extent of rooting was more serious in “Houmairi”, and increased extent of proline content was higher in “Palazzo”.
4. Line 353-356, why did author mention the relationship between chlorophyll and chloroplast? What can this knowledge indicate in this study?
5. Authors explained the NBI at the start of this paragraph, it is better to explain something about this parameter of this study in this paragraph. What did this parameter indicate in this study?
6. Line 366-390, how dose salinity causes the loss of cell membrane integrity? Authors discussed a lot about K+ and Na+ in plant tissues under salinity, however, there were not any data about two ions in this study. I suggest authors rearrange this paragraph.
Minor comments
1. Line 113, change Tuckey test to Tukey’s test.
2. Please change Table 2 and Table 4 as the same form as Table 1, change the capital letters “A”, “B”, “AB”, “BC”, “C” to lower case letters in the Table 1.
3. Line 138, 0.41 was not the same with the data of shoots length of Palazzo in control in Table 2.
4. Line 186, please clearly mentioned which parameter about “this parameter”, because authors mentioned two parameters (root number and root length) in the previous paragraph.
5. Please add lower case letters for the mean values in the Table 3 and checked the description in the section 3.2 followed mean values.
6. In the Table 5, please clearly check the mean values and added correct lowercase letters. For instance, the values 0.39±0.06a, should be 0.39±0.06bc or 0.39±0.06c. Please added lowercase letters for mean values of data in Houmairi, and the significance of ANOVA of Genotype and Genotype´Treatment.
7. Line 263, delete “the” after “Finally,”.
8. Unify the “ns” to “NS” in all tables.
9. In the Table 6, change “lekage” to leakage, and add unit of each parameter.
Comments on the Quality of English LanguageModerate editing of English language required
Author Response
Author's Reply to the Review Report (Reviewer 3)
Comments and Suggestions for Authors
Dear Authors
The manuscript horticulturae-2675561 titled “Application of the encapsulation technology: in vitro screening of two Ficus carica L. genotypes under different NaCl concentrations” has done a vitro work to determine the different tolerance of two Fig tree cultivars. It was interesting, however, I still have some major and minor comments as follow.
Reply: The authors thank the reviewer for his assessment and appreciate his acknowledgement that the manuscript can be improved. We have attempted to implement his recommendations for this purpose.
____________________________________________________________________________________
Major comments
Comment 1: Authors didn’t carefully arrange the data in tables and did many mistakes in tables as I mentioned in the minor comments, please carefully rearrange them and double check them.
Reply 1: The authors revised and restructured the tables based on reviewer suggestions, as reported within the manuscript.
______________________________________________________________________________________
Comment 2. I suggest authors clearly check the results section and check all the description in English by native English speaker.
Reply 2: The results section was revised as correctly suggested by the reviewer, about the presentation of the data in the restructured tables. Furthermore, a language check was conducted.
______________________________________________________________________________________
Comment 3: In line 342-343 and line 442, it is difficult to conclude that “Houmairi” in more tolerant than “Palazzo”, because the decreased extent of rooting was more serious in “Houmairi”, and increased extent of proline content was higher in “Palazzo”.
Reply 3: The data in Table 2 suggests that salinity has a detrimental impact on the quantity and size of roots produced. Specifically, the Houmairi genotype exhibited lower values, particularly in the 150 and 200 mM treatments, while the Palazzo genotype displayed reductions as early as the 50 mM treatment level.
______________________________________________________________________________________
Comment 4: Line 353-356, why did author mention the relationship between chlorophyll and chloroplast? What can this knowledge indicate in this study?
Reply 4: As part of this study, we evaluated the chlorophyll content and NBI index. These two factors are important indicators of the photosynthesis rate and nitrogen content, which are crucial for plant growth. There is significant evidence to suggest that salinity affects both of these parameters. To support our findings, we have cited some relevant works.
______________________________________________________________________________________
Comment 5: Authors explained the NBI at the start of this paragraph, it is better to explain something about this parameter of this study in this paragraph. What did this parameter indicate in this study?
Reply 5: As the NBI (Nitrogen Balance Index) parameter is an index that considers the relationship between primary and secondary metabolism, it depends on the content of chlorophylls, anthocyanins and flavanols. It represents a sensitive indicator of plant nitrogen N status because when plants are subjected to low availability of nitrogen, the content of (Chl) decreases while epidermal polyphenolics (Phen) increase; so, it represents a useful method to estimate stress-related metabolism parameters.
___________________________________________________________________________________
Comment 6: Line 366-390, how dose salinity causes the loss of cell membrane integrity? Authors discussed a lot about K+ and Na+ in plant tissues under salinity, however, there were not any data about two ions in this study. I suggest authors rearrange this paragraph.
Reply 6: The authors agreed to reorganize this paragraph to enhance the comprehensiveness of this section in the discussion of the results.
______________________________________________________________________________________
Minor comments
Comment 1: Line 113, change Tuckey test to Tukey’s test.
Reply 1: Done.
______________________________________________________________________________________
Comment 2: Please change Table 2 and Table 4 as the same form as Table 1, change the capital letters “A”, “B”, “AB”, “BC”, “C” to lower case letters in the Table 1.
Reply 2: The tables were restructured based on reviewers' identified suggestions.
______________________________________________________________________________________
Comment 3: Line 138, 0.41 was not the same with the data of shoots length of Palazzo in control in Table 2.
Reply 3: Corrected
__________________________________________________________________________________
Comment 4: Line 186, please clearly mentioned which parameter about “this parameter”, because authors mentioned two parameters (root number and root length) in the previous paragraph.
Reply 4: Done
______________________________________________________________________________________
Comment 5: Please add lower case letters for the mean values in the Table 3 and checked the description in the section 3.2 followed mean values.
Reply 5: Table 3 was revised based on reviewers' requests to improve data understanding and correspondence with results commentary.
______________________________________________________________________________________
Comment 6: In the Table 5, please clearly check the mean values and added correct lowercase letters. For instance, the values 0.39±0.06a, should be 0.39±0.06bc or 0.39±0.06c. Please added lowercase letters for mean values of data in Houmairi, and the significance of ANOVA of Genotype and Genotype´Treatment.
Reply 6: After considering the suggestions of the reviewers, the authors revised Table 5 to present the data more accurately.
__________________________________________________________________________________
Comment 7: Line 263, delete “the” after “Finally,”.
Reply 7: Done
______________________________________________________________________________________
Comment 8: Unify the “ns” to “NS” in all tables.
Reply 8: Done
______________________________________________________________________________________
Comment 9: In the Table 6, change “lekage” to leakage, and add unit of each parameter.
Reply 9: Done
Round 2
Reviewer 1 Report
Comments and Suggestions for Authors
The authors have made all the necessary changes.
Reviewer 2 Report
Comments and Suggestions for Authors
The authors made several changes that greatly improve the quality of the manuscript. Given this, I believe it can be considered for publication.
Reviewer 3 Report
Comments and Suggestions for Authors
Dear authors,
The manuscript has been revised well.